# A Mobile Device for Monitoring the Biological Purity of Air and Liquid Samples

**DOI:** 10.3390/s21103570

**Published:** 2021-05-20

**Authors:** Tomasz Sikora, Karolina Morawska, Wiesław Lisowski

**Affiliations:** Military Institute of Chemistry and Radiometry, 00-910 Warszawa, Poland; k.morawska@wichir.waw.pl (K.M.); w.lisowski@wichir.waw.pl (W.L.)

**Keywords:** detector for bacteria for fungi, detection of microorganisms, biological air purity

## Abstract

A detector for identifying potential bacterial hazards in the air was designed and created in the Military Institute of Chemistry and Radiometry in the framework of the project FLORABO. The presence of fungi and bacteria in the air can affect the health of people in a given room. The need to control the amount of microorganisms, both in terms of quantity and quality, applies to both hospitals and offices. The device is based on the fluorescence spectroscopy analysis of the sample and then these results were compared to the resulting spectrogram database, which includes the standard curves obtained in the laboratory for selected bacteria. The measurements provide information about the presence, the type, and the approximate concentration of bacteria in the sample. The spectra were collected at different excitation wavelengths, and the waveforms are specific for each of the strains. It also takes under analysis the signal intensities of the different spectra (not only shape a maximum of the peak) so that the concentration of bacteria in the sample being tested can be determined. The device was tested in the laboratory with concentrations ranging from 10 to 10^8^ cells/mL. Additionally, the detector can distinguish between the vegetative forms of spores of the bacteria.

## 1. Introduction

Microorganisms are part of the world we live in. We must coexist and accept the consequences of such an ecosystem. The organisms are oligotrophic in nature and have the ability to spread and easily reproduce. This resilience has allowed them to survive and even thrive in almost all environments [1]. A small amount of organic matter is enough for them to live.

Among the huge number of microbial species, there are some that may pose a health risk. Bacteria and fungi are two of the five microbial agents that can be harmful to humans and thus cause specific diseases. Each one has its own specificity and its own individual structure, the methods of treatment for the ailments they cause are different, and their mode of transmission to humans may differ.

Infections caused by bacteria and fungi have always been a serious medical problem. The most dangerous of them is systemic infection, i.e., sepsis. Paradoxically, despite the development of medical knowledge and the introduction of newer therapeutic procedures, the incidence of sepsis is increasing. It is estimated that over 30 million people worldwide suffer from sepsis each year, with a potential 6 million yearly deaths [2]. It is therefore a more frequent cause of death than cancer. The most important and most difficult problem in treatment is the effective diagnosis of factors causing the systemic inflammatory response. Currently, blood cultures on special media are the diagnostic standard. The disadvantage of this method is that it is time-consuming and the sensitivity is low. In recent years, the development of many new diagnostic methods has been observed. They are based on various molecular techniques, which, unlike traditional diagnostic methods, allow us to obtain accurate and highly specific results in a short time [3]. The diagnostic methods that enable the effective, precise, and quick diagnosis of bloodstream infections include molecular biological techniques such as PCR (Polymerase Chain Reaction) [4] and DNA hybridization [5]. Unfortunately, the PCR reaction has some defects. In samples taken from the environment, there are many substances, namely DNA polymerase inhibitors (e.g., polysaccharides and polyphenols), which can inhibit DNA duplication and give false negative results [6]. Research has been conducted to develop innovative diagnostic methods allowing for the imaging of microbial cells in a blood sample using a fluorescent microscope [7,8]. The main advantages of the method include covering the entire panel of bacterial and fungal microorganisms, differentiating into Gram-negative bacteria, Gram-positive bacteria, yeast-like and mould fungi, without typing specific species, and simultaneous amplification of at least two DNA sequences, which allows for the simultaneous detection of bacteria and fungi. Information on this subject makes it possible to apply more precise treatment with antibiotics. The sensitivity of these methods far exceeds that of the breeding methods [9].

Another area where microorganisms are harmful is with regards to the storage of library materials [10]. The primary factor contributing to the destruction of books in libraries is moisture. It is the most important factor influencing the development of microorganisms responsible for the destruction of library collections. The microorganisms that destroy book collections include bacteria and, above all, fungi [11]. The expanding fungus colony begins to produce spores, which, scattered around the colony, can germinate and form more colonies, further damaging the book. Molecular methods, mainly PCR, are also used to identify the communities of fungi colonizing paper samples of different composition and age [12].

A similar problem can be encountered in the field of museology. Both museum rooms and monument conservation studios are specific micro-environments due to their strictly defined functional character. In these types of interiors, there may be increased contamination with harmful biological agents, which are usually microorganisms that grow on the collected objects or on the surfaces of the rooms [13]. Works of art, due to the origin of the material from which they are made and their diversity, can be freely used as a rich source of nutrients ensuring favourable conditions for the development of various groups of bacteria and moulds [14]. The fact that they spread easily by air means that, under favourable microclimatic conditions, they can not only infect the crops, and consequently initiate the process of biodeterioration, but also have a negative impact on the health of people staying or working in such rooms, breathing in microbiologically contaminated air, because they emit harmful structures and biologically active substances such as allergens, mycotoxins and endotoxins.

The presence of fungi and bacteria in the air can also affect people’s health in office buildings. Installations such as ventilation systems or air conditioning create good conditions for the development of microbes; therefore, their quantity and quality should be monitored. Currently, methods are used to take air samples and then transport them to the laboratory, where they are multiplied and marked. The entire procedure may take up to two weeks. The use of devices capable of giving a real-time result or on-site result would significantly affect the speed of reaction changes.

The problem with air cleanliness is not limited to living quarters or offices. It is also very important to control the quantity (and quality) of microbes in hospital rooms, where the so-called nosocomial infections can be found. Currently, there is access to a wide range of tests using immunological, biochemical, bioluminescence and nucleic acid procedures to identify biological agents. So far, no ideal platform has been developed, but many technologies are capable of detecting microbes.

Currently, microbiological analyses of samples are carried out by cultures on selective media or the use of techniques for detecting characteristic chemical substances, antigens or specific DNA sequences. The latter offer quick and reliable identification of microbes [15,16]. Compared to systems based on antigen–antibody interactions, they show higher sensitivity of the determinations. The application of the PCR technique enables the detection of up to 10 microorganisms [17,18,19]. A serious limitation of this method is the necessity to use pure samples with an appropriate content of nucleic acids. The use of tests based on the immune response requires a series of simultaneous measurements. This is due to the specificity of antibodies, usually directed towards one antigen.

One method that can be used to detect bacteria is fluorescence. This phenomenon is based on the emission of radiation, which is not the glow of bodies heated to high temperatures. For fluorescence to occur, an exciting factor is needed. The name will depend on its nature, e.g.,: electro-, bio-, chem- or photoluminescence. The amount of light emission generated in the fluorescence reaction is measured in the so-called RLU (Relative Light Unit) units. The method allows for a very precise and quick measurement of the tested compounds. In the 1960s, this technology was used by NASA to detect various forms of life on other planets. Currently, fluorescence is the basis for the operation of a variety of sensors, indicators and detectors. It has also found application in determining the concentrations of potentially carcinogenic compounds, aromatic hydrocarbons and heavy metals. It is effective for conducting research on the microbiological contamination of water, food and drugs. Fluorometry is characterized by very high accuracy of measurements and high sensitivity with a wide dynamic range, while maintaining a relatively low cost of the apparatus. However, the most widely used examples of using fluorescence were in the control of the hygienic condition of production surfaces and the assessment of the effectiveness of washing and disinfection processes.

The invention of the fluorescence spectroscopy tool significantly influenced the development of biological sciences. This technique is widely used, among others, in determining cell structures, studying biochemical reactions and in laboratory, medical and microbiological diagnostics. Modern methods of identifying microorganisms based on molecular biological methods allow for obtaining reliable results. On the other hand, these techniques are complex and time-consuming, and qualified personnel are required to analyse the samples. Fluorophores present in microorganisms exhibit characteristic emission bands under the influence of UV radiation.

The first step in developing devices to detect potential biohazards is creating a fluorescence database for various materials. Excitation–emission matrices are a kind of fingerprint of biological molecules. Only on this basis is it possible to optimally select the excitation and emission wavelengths and to design a biological aerosol detection system.

The proposed solution uses a spectroscopic technique based on excited fluorescence. Samples of air or water would be transported to a special chamber where the measurement process would take place. The obtained information would be transferred to a computer and analysed. The fluorescence of bacteria depends on the organic compounds which are part of the cell. Analysis of the spectra in the UV-Vis-IR [18] region can provide a large amount of information which forms the basis for the characterization and identification of the bacteria [19,20,21]. The fluorescence method is based on collecting information about the wavelength of excitation light and recording the emission spectra [22,23,24]. This allows for the development of an excitation–emission dependency matrix.

The basis of its operation is the phenomenon of excited fluorescence. The sample is introduced into the measuring chamber, which is then illuminated with the light of selected wavelengths: 240 nm, 255 nm, 270 nm and 285 nm. Then, depending on the type of research carried out, light is collected after passing through the sample or the light emitted after excitation of the biological material. The spectrum obtained is the basis for the identification of the tested fungi or bacteria. The samples are tested as a solution.

## 2. Materials and Methods

### 2.1. Preparation of Microbial Strains

The microbial strains were prepared and delivered by the team from the Provincial Infectious Hospital in Warsaw in the Laboratory of Molecular Diagnostics. After culturing, the bacterial pellet was suspended in a sterile NaCl solution at a concentration of 0.9%. The optical density of the bacterial vegetative cell suspensions was determined using a densitometer. The density of the suspensions was adjusted to a value of 0.5 on the McFarland scale, which corresponded to a concentration of about 10^8^ cells/mL. The exact cell concentration was determined by plating in Petri dishes. Successive standard solutions were obtained by diluting the stock solution. The device tests were performed with the bacteria and fungi listed in Table 1.

### 2.2. Parameters of the Optical Set of Bacteria and Fungi Detector

The device consists of (Figure 1): optical spectrometer, set of optical sliding filter, set of optical fibers and measuring probe were purchased from Ocean Optics/Ocean Insight, Orlando, FL, USA. The set of UV diodes was purchased from Set Inc. 110 Atlas Ct, Columbia, SC, USA. A linear actuator was purchased from Igus^®^ GmbH Spicher Str. 1a 51147 Köln, Germany. A single-chip microcomputer was purchased from Raspberry Pi. The measuring chamber, set of optical probe adjustment, electronic control modules, chassis, panels, etc. were made according to personal projects and as personal constructions.

The optical probe QR400-7-SR has a wavelength range of 200 nm–1.1 μm. The fiber core size is 400 µm, and the probe ferrule diameter is 6.35 mm (1/4″). Optic fiber length is approximately 2 m. The applied reflection probe has a ring with additional fibers of optical fibers to illuminate the sample, which does not change the fact of using the central wire to collect the spectrum. LEDs Electric power is approximately 150 mW and optical power is approximately 340 μW ÷ 400 μW depending on the wavelength.

In the receiving path, between the chamber and the measuring probe, there is a high-pass, interference optical LVF-UV filter with a spectral transmittance in the range of 230–500 nm. The basis of the filter is a quartz glass plate on which optical coatings, made with thin-film techniques, has been deposited.

The filter is moved by a linear actuator. This design is based on a low-profile, flat, linear guide system. A trolley made of high-performance self-lubricating polymer material moves along the rail. An integral part of the drive is a stepper motor with a resolution of 200 steps per revolution (1.8°). The applied lead screw in the metric system provides a single “step” of filter movement with a length of 4 µm. This allows for precise filter movement in the optical path. The engine is controlled by a dedicated logic. The detection element of the system is a reflection probe. It is mounted in the adjustment unit, which enables the positioning of the probe in three axes. The adjustment range is: X and Y axes +/− 1 mm (diameter), Z axis 4 mm. A receiving optical fiber (400 μm) is attached to the optical spectrometer. Maya2000 Pro spectrometer parameters: dimensions 148.6 × 109.2 × 46.4 mm, Hamamatsu S10420 detector (the best option for UV-Vis spectrometry) with spectral spectrum 200–500 nm, useful signal-to-noise ratio approx. 450:1, dynamic range 15000:1, integration time 7.2 ms–5 s.

As a light source, laser diodes with wavelengths are used. Diode ranges: 240, 255, 270, 285 nm.

On the top cover, there is a measuring cuvette socket with a bayonet closure. This structure, after closing the housing, creates a double labyrinth system for external light with the measuring chamber placed on the mounting plate.

As part of the work, the following were created:An application for desktop computers (MayaDesktop).An application for a single-board, mini-computer with a Unix-like operating system based on the Linux kernel (MayaRemote).

The main functions of MayaDesktop are: communicating with MayaRemote to collect measurements with data, displaying graphs with Maya spectrometer data sets, as a collection of measurements for one biological sample using 4 LED irradiation for different filter setting ranges. It enables single or serial measurements of a biological sample. It allows you to manage the collected data in the database and to create a library with reference charts.

MayaRemote allows the UART communication protocol to be established between the MayaRemote application and the microcontroller, sending messages according to a predetermined communication protocol that will control the four LEDs and the filter, collecting biological sample data from a Maya spectrometer.

### 2.3. The Process of Testing Biological Samples

Laboratory tests and studies were performed on the cell suspension. The presented device has been prepared for standard spectrophotometric cuvettes used for liquid testing. However, the device can also be used for air sampling. A bacterial aerosol was prepared in the BSL chamber and then collected by the air collecting system of own design. The principle is to transfer the airborne particles into a liquid sample (Figure 2). Basically, air is aspirated and drawn into the cone forming a vortex. Bioparticles are centrifuged into the wall and separated from the air—contaminants are trapped in the solution. After the process is complete, the liquid sample is ready and can be transferred for further analysis by use of the presented device. From this point, the sample is treated in the same way as the liquid sample. The cells’ suspension sample was transferred to a cuvette and measured.

The effectiveness of air sampling with this method was confirmed by culturing the obtained suspensions. Further tests of the device were carried out with the use of liquid samples—cell suspension. It resulted from the necessity to decontaminate the BSL chamber after each aerosol application, and, for safety reasons, uncontrolled contamination of the laboratory room.

The solution of tested microorganisms was placed in the UV-Vis spectrophotometric cuvette in the amount of 1 mL to 2 mL, and then transferred to the measuring chamber of the detector. The sample is excited at different wavelengths, and then the spectrum of radiation emitted by the tested material is recorded. The optical signal is directed to a fiber optic probe, transferred by optic fibers to the spectrometer connected with a built in computer.

The measurement is based on excitation of the sample using four different wavelengths. Fragments of the excitation spectrum are collected. These fragments are cut out of the obtained spectrum by appropriate setting of optical filters. The arrangement of two filters, high-pass and low-pass, allows for separating certain fragments from the emission spectrum. They are analysed separately. This allows for the removal of the non-specific glow or the decomposition of overlapping peaks. For each length of the excitation light, an emission spectrum was obtained, which was passed through optical filters. In addition, 64 different filter settings were tested for each excitation light wavelength, resulting in up to 256 spectral fragments per measurement. The number of filter settings was reduced, but at least 16 different settings were still applied.

For each fragment, a peak was selected for which the position and intensity were determined. In this way, a set of points was created, described by the length of the excitation light, the position, and intensity of a given peak. This information has been saved in a database. The results of the measurements made are compared with those recorded for the reference samples and the degree of similarity has determined, which is the basis of identification.

Collected fluorescence emission spectrum at each of the wavelengths of the UV LEDs compared with the reference spectra stored in the database. Standard spectra were obtained during laboratory tests at the Provincial Infectious Hospital in Warsaw in the Laboratory of Molecular Diagnostics with the use of live bacteria. The complete list of microorganisms that were tested is provided in Table 1. The range of tested concentrations ranged from 10 to 10^8^ cells/mL.

## 3. Results

Based on the data collected during the research, a database was created containing sets of spectra for all fungi and bacteria presented in Table 1. In order to create the identification algorithm, the emission spectrum was divided into fragments. Each of the analysed fragments was subjected to mathematical processing to check whether there were peaks in the spectrum. If overlapping peaks were present, attempts were made to decompose them. Such peaks in the analysed spectral fragments are called characteristic points. It is worth adding that the identification does not take place after detecting a single peak. Their positions, intensity and their relation to other peaks in the remaining fragments are measured. The range of tested concentrations ranged from 10 to 10^8^ cells/mL. The graph in Figure 3 shows fragments of spectra obtained for *E. coli* after the use of optical filters. Each curve corresponds to a different concentration of bacterial cells in the test sample. The excitation took place at a wavelength of 240 nm. Figure 4, Figure 5 and Figure 6 also show fragments of spectra obtained for *E. coli*, but at the wavelength of excitation light of 255, 270 and 285 nm.

As shown in Figure 3, Figure 4, Figure 5 and Figure 6, the position of the emission peak occurs at the same wavelength for each measurement and only the peak intensity changes, which increases with increasing cell concentration in the test solution. This is evidence that this peak comes from bacteria and is directly related to their amount. Such changes in signal intensity were observed for all tested fungi and bacteria. The developed algorithm is based on the analysis of the excitation spectra and the selection of characteristic points. Then, such analysis is carried out for all excitation wavelengths and filter settings. A matrix of results related to the excitation wavelength, the position of the optical filter (the cut and transmitted lengths of the emitted light), and the position and intensity of the peaks in the emission spectrum are created. Relative relationships for each setting for selected ranges are analysed. The resulting matrix of results describes the bacteria or fungi tested. The identification takes place by comparing the characteristic points in the spectrum obtained for the tested sample with the reference spectra stored in the database. Figure 7 shows three series of measurements for *E. coli* using optimized optical path configuration and applying a differential signal analysis algorithm. The graph shows the intensity of the selected peak in the spectrum as a function of the concentration of bacterial cells. The following series show the optimization of the positioning of the lenses in relation to the optical probe.

The next figures show the influence of the measurement time on the detection capabilities. Figure 8 was performed with the measurement times: 30 ms (series 5 and 6), 60 ms (series 1 and 2) and 100 ms (series 3 and 4).

Although higher signal intensities were obtained at 100 ms, some resolution was lost. At 60 ms, lower peak intensities were obtained, but they were better distinguished.

The full measurement cycle required illumination for each excitation length wave and subsequent settings of the optical filter. When using a longer excitation time, the obtained spectra became blurry. After reducing the time to 60 ms, the problem was resolved. It is assumed that this effect was related to local heating of the sample and diffusion with convective movements. Each of the recorded peaks is presented on the plot as a point. As you can see, while maintaining the same measurement parameters, the obtained curves maintain high compliance. However, when changing the measurement time, the differences that appear are already significant. After obtaining the emission-absorption characteristics for all microorganisms, subsequent versions of the algorithms were developed, and the research on coded samples was started.

Algorithms were created to perform mathematical operations and subsequently the obtained results were compiled in numerical form. Identification according to algorithms was carried out by comparing the results obtained for the tested sample with the sets of results stored in the database. The difference between the algorithms is that they give a different weight to the selected sets. This was intended to help in the analysis of less distinguished peaks. The A1 algorithm scored clearly distinguished peaks higher, and the less distinguished peaks were treated as Supplementary data of less significance. The A2 algorithm was created to assist in the analysis of the latter category of less distinguished peaks. As a result, the algorithm has become more susceptible to distortion of the emission spectrum.

The concentration of the bacteria in the suspension was assessed using a densimeter in McFarland units. As a result, a calibration curve was created that allowed concentration estimation. The device only gives an estimate of the amount and not an exact value. The signal intensity measurement was used as the basis for quantification. Measurement in a mixture is possible if the concentrations of both bacteria are similar; otherwise, it becomes impossible to distinguish them because the signals may overlap and become indistinguishable. In the example described, the mixture was tested in a ratio of 1:1. Three identification attempts were made.

Attempt 1

Solutions of different concentrations were prepared for the two different bacteria *Enterococcus faecalis* and *Enterococcus faecium*. The samples were prepared by a team from the Infectious Hospital in Warsaw. The selected bacteria were chosen as an example of bacteria that are difficult to distinguish from each other due to similar biochemical properties. In the analysis, the detection algorithm denoted as A1 was used. Results: bacteria were correctly identified, and their concentration was determined for concentrations from 10^3^ to 10^8^ cells/mL using the excitation time t = 60 ms.

Attempt 2

Solutions of 16 different bacteria/fungi with equal concentrations were prepared. The results are presented in Table 2.

Eleven out of 16 samples were correctly identified using the algorithm denoted as A1. Two samples were misidentified (false positives). In one sample, no bacteria were detected—false alarm was negative. All Gram-positive bacteria and fungi were correctly identified. All misidentified samples contained Gram-negative bacteria.

Attempt 3

As in attempt 2, solutions of 12 bacteria/fungi with similar concentrations were prepared. However, in this attempt, the detection algorithm designated A2 was used. The results are presented in Table 3.

The second algorithm was less effective and only recognized 6 out of 12 samples. One sample was incorrectly identified—false positive-positive, and three samples were not identified and marked as free of bacteria—false positive-negative. Again, Gram-positive bacteria and fungi were identified positively.

## 4. Conclusions

The conducted research indicates the potential possibilities of using the detector as a device for quick detection and identification of fungi and bacteria. At the same time, the diversity of the characteristics of fluorescent biological preparations requires a more detailed analysis of the spectra supported by advanced statistical methods for the detection and recognition of biological hazards. At this stage, it is possible to identify between fungi and Gram-positive bacteria or mixtures with similar concentrations of microorganisms (two species). Further work will be focused on testing multi-component mixtures and mixtures with a large difference in concentration. The device can detect Gram-positive fungi and bacteria. Using the A1 detection algorithm, all bacteria belonging to the group mentioned were identified positively, and errors occurred with samples containing Gram-negative bacteria. One can also positively interpret the fact that the device compared wrongly identified samples only with other Gram-negative bacteria and never with Gram-positive ones. The A2 detection algorithm was found to be less accurate but confirmed the device’s ability to identify groups of fungi and Gram-positive bacteria. False identification resulted from samples that were too close in similarity which were obtained during the comparison of the tested samples with reference data.

The problem concerns Gram-negative bacteria. We assume that the thickness and the composition of the cellular wall are of major importance here. The problem needs to be analysed in terms of differences between Gram-negative bacteria and their composition because we cannot improve the algorithm from the software level.

The undoubted advantage of this device is the ability to obtain results in virtually real time without the need to transfer samples to the laboratory. The great challenge is to increase the selectivity of the method. Therefore, work is carried out concurrently on the simultaneous measurement of additional parameters and the introduction of an additional measurement channel.

Further work will focus on increasing the device’s ability to detect and identify Gram-negative bacteria.

In summary, the device is characterized by:-the ability to adapt the database to the user’s requirements—only the microbes that are required will be recognized, especially when used in monitoring mode, where the presence of specific threats is expected. Moreover, it shows the speed of determination;-the ability to adapt the measurement cycle to specific determinations—general or screening measurements;-possibility to automate the detection process;-the database can be built in or external, wirelessly connected to the server.

An additional advantage is the portability of the device—small size and weight, so it can be used anywhere. Further work on the device is planned in order to improve it.

## Figures and Tables

**Figure 1 sensors-21-03570-f001:**
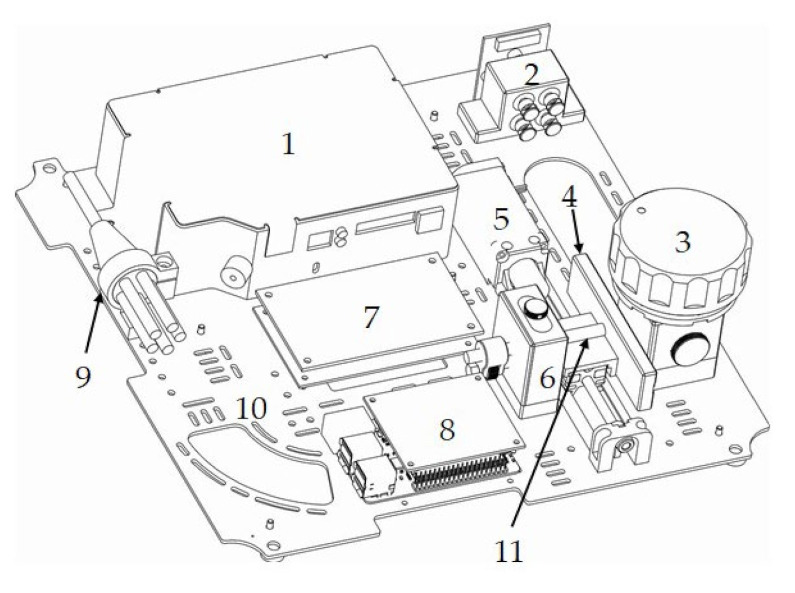
View of the assembled device:.1—optical spectrometer, 2—set of UV diodes, 3—measuring chamber, 4—optical filter, 5—linear actuator, 6—optical probe adjustment unit, 7—electronic control modules, 8—single-chip microcomputer, 9—set of optical fibres, 10—construction plate, 11—probe.

**Figure 2 sensors-21-03570-f002:**
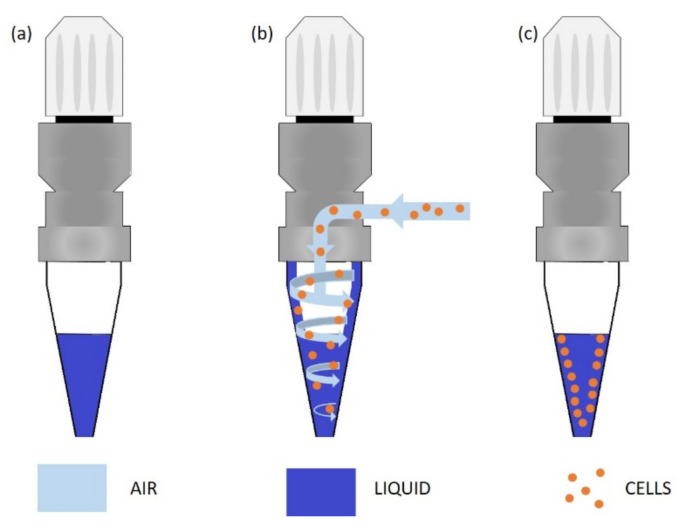
Cone with 0.9% NaCl solution (**a**), aspiration of the air stream by vortex inside the cone (**b**), cells are trapped in the liquid and separated from air (**c**). The liquid sample with cells is ready for analysis.

**Figure 3 sensors-21-03570-f003:**
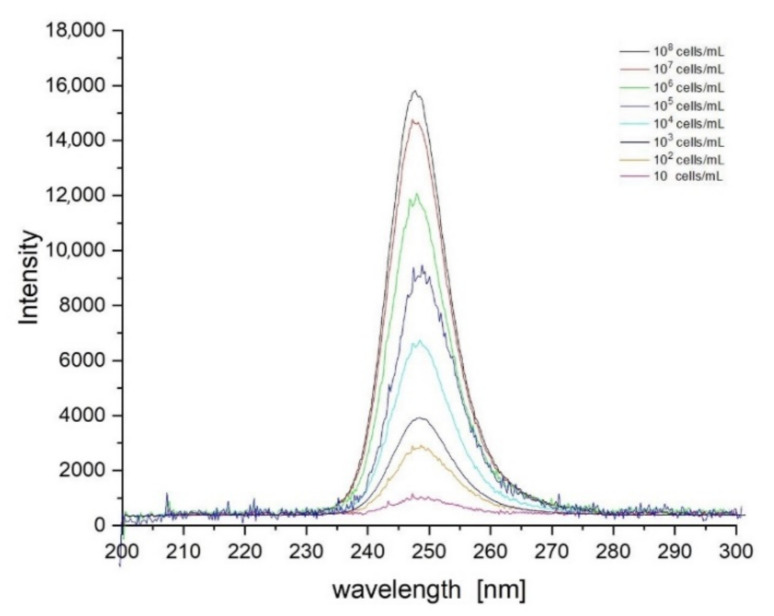
Set of spectra obtained for various concentrations of E. coli at 240 nm excitation. Optical filters were set to cut off illumination of scattered excitation light.

**Figure 4 sensors-21-03570-f004:**
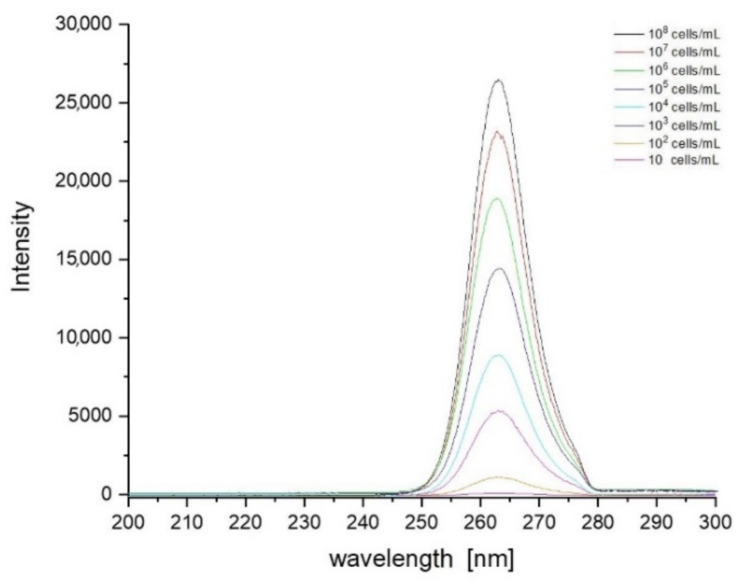
Set of spectra obtained for various concentrations of *E. coli* at 255 nm excitation. Optical filters were set to cut off illumination of scattered excitation light.

**Figure 5 sensors-21-03570-f005:**
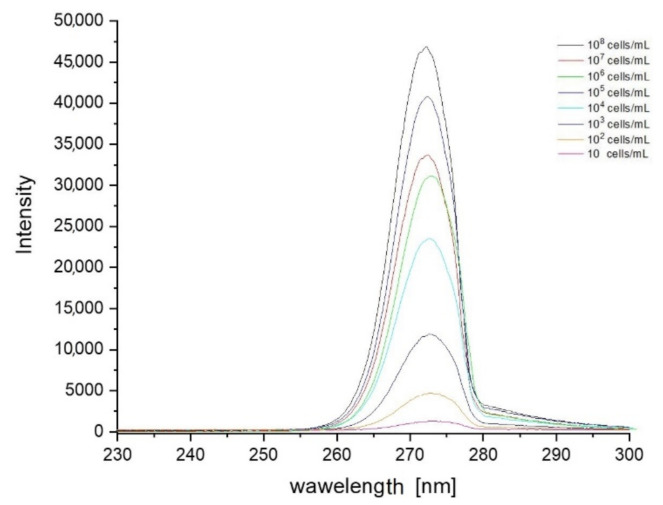
Set of spectra obtained for various concentrations of *E. coli* at 270 nm excitation. The peak is misshapen on the right side (longer wavelengths) due to proximity of the maximum of the emission spectra and excitation wavelength.

**Figure 6 sensors-21-03570-f006:**
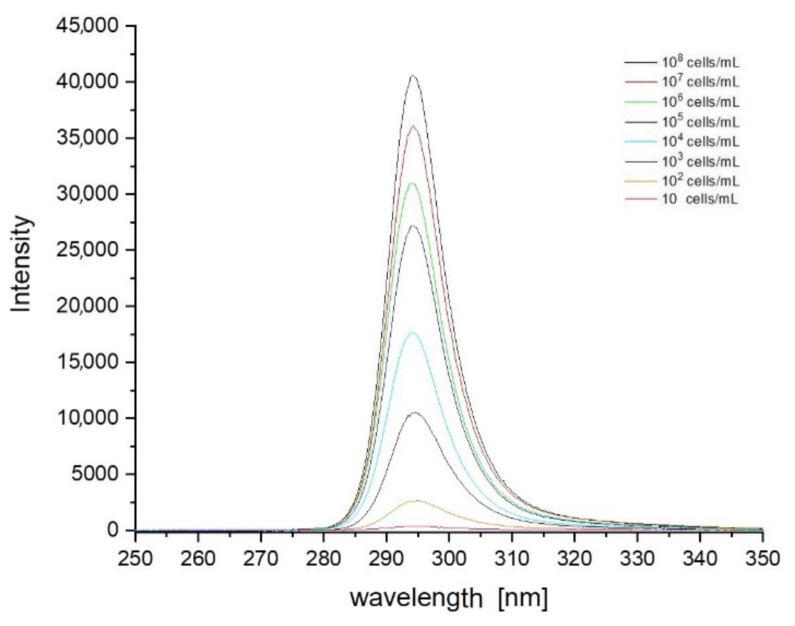
Set of spectra obtained for various concentrations of *E. coli* at 285 nm excitation. Optical filters were set to cut off illumination of scattered excitation light.

**Figure 7 sensors-21-03570-f007:**
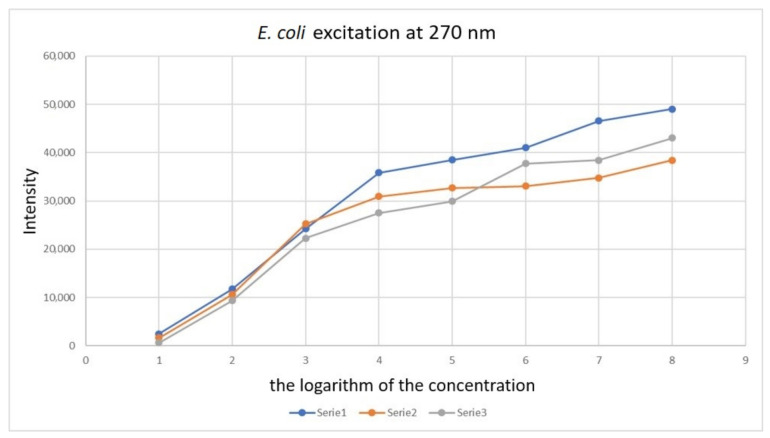
Graph showing changes in the peak maximum recorded by excitation with a wavelength of 270 nm depending on the concentration of *E. coli* bacteria using the appropriate configuration of the optical path.

**Figure 8 sensors-21-03570-f008:**
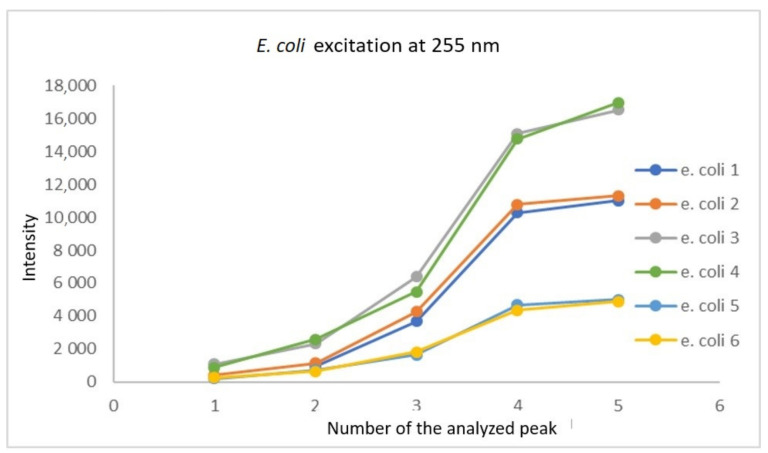
Influence of the measurement time on the intensity of the recorded peaks for *E. coli* at 255 nm excitation wavelength.

**Table 1 sensors-21-03570-t001:** Microorganisms tested in the framework of the project.

No.	Microorganisms	Description
Fungi		
1	*Aspergillus* spp.	species of fungi from the family *Trichocomaceae*
2	*Candida albicans*	a species of fungi classified as yeasts, non-enveloped
3	*Candida glabrata*	a species of fungi classified as yeasts, non-enveloped
4	*Candida krusei*	hospital pathogen
5	*Candida tropicalis*	a species of fungi belonging to the yeast family, one of the most virulent of the genus *Candida*
6	*Cryptococcus neoformans*	species of fungi from the *Tremellacea*e family
**No.**	**Microorganisms**	**Description**
**Bacteria**		
1	*Acinetobacter* spp. *(Acinetobacter baumannii)*	a species of Gram-negative, non-fermenting bacteria, is a common nosocomial infection in patients requiring a large number of invasive procedures
2	*Corynebacterium* spp.	aerobic or facultative anaerobic Gram-positive bacteria that can spore
3	*Citrobacter* spp. *(**Citrobacter koseri)*	bacteria belonging to the Enterobacteriaceae family, accepting large, non-enveloped, non-spore Gram-negative, non-spore-forming bacilli
4	*Enterococcus faecalis*	*Faecal streptococcus*, Gram-positive bacterium belonging to *Enterococci*
5	*Enterococcus faecium*	Gram-positive *enterococci*, physiologically present in the human gastrointestinal tract
6	*Escherichia coli*	coliform, Gram-negative, relatively anaerobic bacteria belonging to the *Enterobacteriaceae* family
7	*Klebsiella pneumoniae*	capsular, non-spore, cystless gram-negative bacterium
8	*Klebsiella oxytoca*	Gram-negative relatively anaerobic rod belonging to *Enterobacteriaceae*
9	*Klebsiella pneumoniae kpc+*	enterobacteria that produces KPC carbapenemases (*Klebsiella pneumoniae carbapenemase*)
10	*Morganella morganii*	Gram-negative bacteria, classified as a normal bacterial flora
11	*Pseudomonas aeruginosa*	blue pus rod, Gram-negative bacteria, resistant to antibiotics, causes nosocomial infections
12	*Proteus mirabilis*	ciliated bacteria, Gram-negative, in humans is a component of the physiological flora of the digestive system
13	*Providencia rettgeri*	Gram-negative bacteria
14	*Salmonella* spp*. (**salmonella typhi salmonella enteritidis)*	a genus of bacteria from the *Enterobacteriaceae* family, grouping Gram-negative or anaerobic rods
15	*Serratia marcescens*	ciliated, non-spore-forming, non-enveloped, Gram-negative bacteria
16	*Staphylococcus aureus*	Gram-positive bacteria, found in the nasopharynx and on the skin of humans and animals, does not produce spores
17	*Staphylococcus epidermidis*	a species of Gram-positive bacteria, belonging to the genus coagulase-negative *Staphylococci* (CoNS)
18	*Staphylococcus haemolyticus*	Gram-positive, relatively anaerobic, non-sporulating bacteria, belonging to the genus coagulase negative *Staphylococci* (CoNS)
19	*Staphylococcus saprophyticus*	Gram-positive bacteria belonging to the genus *Staphylococci*
20	*Stenotrophomonas maltophilia*	Gram-negative, non-sporulating bacteria belonging to the group of non-fermenting rods
21	*Streptococcus pneumoniae*	*Pneumonia diphtheria*, aerobic bacterium, Gram-positive, belongs to alpha-hemolytic *Streptococci*
22	*Streptococcus beta-hemolytic,* *Streptococcus pyogenes, Streptococcus agalactiae*	Gram-positive beta-hemolytic *Streptococcus* belongs to cocci
23	*Streptococcus viridans*	bacteria from the group of alpha-hemolytic *Streptococci* and non-hemolytic bacteria

**Table 2 sensors-21-03570-t002:** Test results with coded samples.

No.	Actual Content	Indications of the Detector
1	*Candida albicans*	*Candida albicans*
2	*Citrobacter koseri*	*Escherichia coli*
3	*Enterococcus faecalis*	*Enterococcus faecialis*
4	*Escherichia coli*	*Klebsiella pneumoniae*
5	*Staphylococcus aureus*	*Staphylococcus aureus*
6	*Enterococcus faecium*	*Enterococcus faecium*
7	*Klebsiella pneumoniae*	Data insufficient
8	*Staphylococcus epidermidis*	*Staphylococcus epidermidis*
9	*Aspergillus* spp.	*Aspergillus* spp.
10	*Candida krusei*	*Candida krusei*
11	*Acinetobacter baumannii*	Data insufficient
12	*Streptococcus pyogenes*	*Streptococcus pyogenes*
13	*Staphylococcus saprophyticus*	*Staphylococcus saprophyticus*
14	*Corynebacterium*	*Corynebacterium*
15	*Morganella morganii*	Data insufficient
16	*Cryptococcus neoformans*	*Cryptococcus neoformans*

**Table 3 sensors-21-03570-t003:** Test results with coded samples.

No.	Actual Content	Indications of the Detector
1	*Escherichia coli*	Data insufficient
2	*Candida albicans*	*Candida albicans*
3	*Citrobacter koseri*	Data insufficient
4	*Klebsiella pneumoniae*	*Escherichia coli*
5	*Staphylococcus epidermidis*	*Staphylococcus epidermidis*
6	*Enterococcus faecium*	*Enterococcus faecium*
7	*Staphylococcus aureus*	*Data insufficient*
8	*Enterococcus faecalis*	*Enterococcus faecialis*
9	*Pseudomonas aeruginosa*	Data insufficient
10	*Stenotrophomonas maltophilia*	Data insufficient
11	*Staphylococcus saprophyticus*	*Staphylococcus saprophyticus*
12	*Candida krusei*	*Candida krusei*

## Data Availability

The data presented in this study are available on request from the corresponding author.

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
