# Peer review of "A Mobile Device for Monitoring the Biological Purity of Air and Liquid Samples"

_sensors, 2021, doi:10.3390/s21103570_

Round 1

Reviewer 1 Report

The authors describe the development of an apparatus that , based on excitation-emission matrices, is able to detect bacteria and fungi. Although the utilisation of excitation-emission matrices for this purpose is not new, the described apparatus and the describes results deserves attention. The manuscript has potential but requires improvement. The paper can benefit by proper response to the remarks as listed below.

  1. The title refers to “detection in air”. However, all the results shown deal with suspensions. It could well be that in a further stage the instrument proves to be useful for monitoring biological purity of air but no evidence is given yet. So, the title is not justified and must be changed.
  2. The authors have to screen more recent literature concerning the use of the induced fluorescence of bacteria and fungi for their detection by excitation-emission matrices to provide the reader with a more up to date status of this type of research. E.g. Dartnell LR, Roberts TA, Moore G, Ward JM, Muller J-P (2013) Fluorescence Characterization of Clinically-Important Bacteria. PLoS ONE 8(9):e75270. doi:10.1371/journal.pone.0075270. Additional references can be found.
  3. In the context of the previous remark, the authors should explicitly indicate the novelty or increased performance of their approach/apparatus.
  4. The terms “luminescence” and “fluorescence” are used throughout the manuscript. However, these terms are not equivalent. Fluorescence is a type of luminescence, the latter being more general. Please use a consistent and correct phrasing.
  5. 3, line 129: “fluorescent database for various materials”. Please change into “ fluorescence database for various materials”. The database itself is not fluorescent.
  6. 1: the number “6” is used twice. Has this a special purpose?
  7. Provide details about the companies (city, country) from which the various components are obtained.
  8. An optical probe is mentioned. This is the so called reflection probe. Please provide details about this reflection probe (company, type of probe, …). In general, a reflection probe comprises illumination fibers and a fiber for collecting the light. For the described apparatus the excitation is provided by UV LEDS. Why is a reflection probe used in the described apparatus?
  9. What is the power of power of the UV LEDs?
  10. 6, line 195: “The applied lead screw in the metric system provides a single "step" of filter movement with a length of 4 m”. How to interpret “4 m”? Is a symbol missing?
  11. 7, line 242: The optical signal is collected by a fiber optic probe and recorded by a computer. How can the computer record the optical signal transmitted through the fiber? The phrasing should be adjusted by indicating that the output of the spectrometer is connected to the computer.
  12. 7, line 243: “At least 16 different optical filter settings are used for each excitation light .... In this way, it is possible to cut specific wavelengths from the complex emission spectrum”. How to understand this as the spectrometer contains a particular grid which in combination with a CCD image sensor has a defined spectral range and resolution? Please provide more details about these 16 settings in relation to the corresponding wavelengths ranges.
  13. 7, line 248: “The obtained spectrogram of the excited spectrum as a function of the excitation radiation wavelength…”. I think “excited spectrum “ is a misnomer. The spectrum is not excited, and what is collected is the fluorescence emission spectrum at each of the wavelengths of the UV LEDs. Please correct.
  14. 7, line 256:” For each spectrum, characteristic points were developed, which were used to develop the detection algorithm.” Apart from the fact that I think that the first “developed” in this sentence is not appropriate to indicate what the authors actually mean, the authors should make clear what they call characteristic points.
  15. What is the relevance of Fig. 2 for the reader? Physical dimensions are difficult to estimate.
  16. Figs 3 and 4 should definitely have a wavelength scale. Provide also details on the filter settings.
  17. 3 and 4: what are the actual concentrations used? Provide a legend to translate the colours into concentrations. Please use the same colour coding in the two figures as I think that the same samples are considered for these two figures.
  18. General remark to the figures: provide the necessary information in the figure legend for the convenience of the reader.
  19. What is the nature of the algorithms A1 and A2. What are the specific characteristics of each of the algorithms? The reader is not provided with any information at all about these algorithms. How can the reader then appreciate the different performance of the A1 and A2. The authors must provide information in this respect.
  20. 10, line 316:” Bacteria were correctly identified and their concentration was … “ Please provide details about the calibration method required to translate the detected optical signals into actual concentrations. The statement seems to suggest also that the bacteria in a mixture can be correctly identified. What is the basis for this statement? In other words, provide results related to attempt 1 that substantiate the statement. What was the ratio if their concentrations for each of the total concentrations considered. This is one way to demonstrate the power of the approach.
  21. In the conclusion one can read:” Already at this stage of the research, it can be stated that the test device gives a great opportunity to distinguish between fungi and bacteria.” It is not all clear from the results shown how a discrimination between different fungi or bacteria will feasible. Please justify this statement, or omit this statement.

Reviewer 2 Report

Please find the comments in the attached file.

Author Response

Answers to the reviewer's questions in the attachment 

Round 2

Reviewer 1 Report

The changes by the authors are appreciated.

However, I have some comments to the revised version.

  1. When the title contains “air samples” then evidence (results) must be shown in the manuscript. I suggest that the authors include the results obtained on the air sample (with detailed description how the air sample was obtained). I think many readers will be interested in this. In the current version there is only the indication that the study on air samples was abandoned.
  2. Concerning the number “6”. I apologize for not being clear. The number “6” appears twice in Figure 1.
  3. 6, lines 190-194: Is this to become part of the legend? Some phrases are not sentences as these do not contain a verb. Please correct.
  4. I appreciate the details about the companies as indicated by the authors in the reply. However, the main aspects should be indicated in the manuscript.
  5. The figure on top of p. 7 has no number or legend. Is this figure to be deleted?
  6. The description of the two algorithms A1 and A2 is still vague. I suggest that the authors indicate that they are willing to share the details of the algorithms upon request.
  7. The summary on p. 16 is actually a list of characteristics. However, the items in the list are separated by “.”. Either the authors rewrite the summary as a list according to a), …b)…, c)… , or they write full sentences that contain a verb.
  8. In the text and in the headers of the graph one can read “E. Coli” and “E. coli”. Please make consistent throughout the manuscript.

Reviewer 2 Report

The authors gave responses to the comments but did not reply how they revised the manuscript for some comments, although the manuscript likely improved.

I did not find a final version of the manuscript. But there are still some typos, format errors, and grammar errors in the marked version.

  1. X-axes of Figs. 3-5, "wavelenght, [nm]"---->"wavelength [nm]"
  2. Table 2 and 3, both in Figures and captions of Figs 6 and 7: "E. coli" should be italic and "E." should be in capital.  Serie1---->Series1? And also similar errors for other genus names.
  3.  The last summary paragraph is not writing in a scientific style. The sentences are incomplete or incorrect in grammar. 
